# Revisiting Adversarial Robustness of Classifiers With a Reject Option

**Jiefeng Chen**[*,1] **Jayaram Raghuram**[*, 1] **Jihye Choi,** [1] **Xi Wu,** [2] **Yingyu Liang,** [1] **Somesh Jha** [1]

[1] University of Wisconsin-Madison
[2] Google
{jiefeng; jayaramr; jihye}@cs.wisc.edu, wu.andrew.xi@gmail.com, {yliang; jha}@cs.wisc.edu

## Abstract

Adversarial training of deep neural networks (DNNs) is an important defense mechanism that allows a DNN to be robust to input perturbations, that can otherwise result in predictions errors. Recently, there is a growing interest in learning a classifier with a reject (abstain) option that can be more robust to adversarial perturbations by choosing to not return a prediction on inputs where the classifier may be incorrect. A challenge faced with robust learning of a classifier with reject option is that existing works do not have a mechanism to ensure that (very) small perturbations of the input are *not* rejected, when they can in fact be accepted and correctly classified. We first propose a novel metric – *robust error with rejection* – that extends the standard definition of robust error to include the rejection of small perturbations. The proposed metric has natural connections to the standard robust error (without rejection), as well as the robust error with rejection proposed in a recent work. Motivated by this metric, we propose novel loss functions and a robust training method – *stratified adversarial training with rejection* (SATR) – for a classifier with reject option, where the goal is to accept and correctly-classify small input perturbations, while allowing the rejection of larger input perturbations that cannot be correctly classified. Experiments on well-known image classification DNNs using strong adaptive attack methods validate that SATR can significantly improve the robustness of a classifier with rejection compared to standard adversarial training (with confidence-based rejection) as well as a recently-proposed baseline.

## Introduction

Training robust classifiers in the presence of adversarial inputs is an important problem from the standpoint of designing secure and reliable machine learning systems (Biggio and Roli 2018). Adversarial training and its variations are the most effective methods for learning robust DNN classifiers (Madry et al. 2018; Zhang et al. 2019). However, adversarial training may still not be very effective against adaptive adversarial attacks, or even standard attacks with configurations not observed during training (Athalye, Carlini, and Wagner 2018; Tramèr et al. 2020). Given this limitation, it is important to design classifiers that learn when to reject or abstain from predicting on hard-to-classify inputs. This

---

*These authors contributed equally.

can be especially crucial when it comes to real-world, safety-critical systems such as self-driving cars, where abstaining from prediction is often a much safer alternative to making an incorrect decision.

We focus on the problem of learning a robust classifier with a reject option in the presence of adversarial inputs. The related problem of learning a (non-robust) classifier with a reject option has been studied extensively in the literature (Tax and Duin 2008; Guan et al. 2018; Cortes, DeSalvo, and Mohri 2016; Geifman and El-Yaniv 2019; Charoenphakdee et al. 2021). Recently, a number of works have also addressed the problem of adversarial robustness for a classifier equipped with a reject option (Laidlaw and Feizi 2019; Stutz, Hein, and Schiele 2020; Sheikholeslami, Lotfi, and Kolter 2021; Pang et al. 2021b; Tramèr 2021; Kato, Cui, and Fukuhara 2020). These approaches extend the standard definition of adversarial robustness (robust error) to the setting where the classifier can also reject inputs. In this setting, rejection of a perturbed input is considered to be a valid decision that does not count towards the robust error. However, rejection of a clean input still counts towards the robust error (Tramèr 2021).

A key limitation with this view of the robust error with rejection is that it treats equally the rejection of very small perturbations as well as large perturbations of an input. However, many practical applications (*e.g.*, object detection) may require that small perturbations of an input be handled accurately by the classifier without resorting to rejection. In other words, there could be a higher cost for rejecting small input perturbations, when in-fact the classifier can accept and classify them correctly. Existing methods for training a robust classifier with rejection, such as confidence-calibrated adversarial training (CCAT) (Stutz, Hein, and Schiele 2020), achieve a high robustness by simply rejecting a large fraction of the perturbed inputs (since rejecting perturbed inputs does not contribute to the robust error, no matter the perturbation size). As we validate experimentally, CCAT often has a high rejection rate on even small perturbations of clean inputs, which may not be acceptable in many practical applications.

Motivated by these limitations in existing works, we revisit the problem of adversarial robustness of a classifier with reject option, and make the following contributions:

- We propose a novel metric – *robust error with rejection* – that can provide a fine-grained evaluation of the robust-

ness of a classifier with reject option across a range of perturbation sizes.

- We provide a theoretical analysis of this problem, which motivates the need for learning a robust classifier with rejection that can accept and correctly classify small input perturbations.
- We propose novel loss functions and a robust training method SATR for jointly learning a classifier-detector system (*i.e.*, a classifier with rejection) that are designed to achieve the goal of accepting and correctly classifying small input perturbations, while also selectively rejecting larger input perturbations.

## Related Work

Adversarial robustness of deep learning models has received significant attention in recent years. Many defenses have been proposed and most of them have been broken by strong adaptive attacks (Athalye, Carlini, and Wagner 2018; Tramèr et al. 2020). The most effective approach for improving adversarial robustness is adversarial training (Madry et al. 2018; Zhang et al. 2019). However, adversarial training still cannot achieve very good robustness on complex datasets, and often there is a large generalization gap in the robustness (Tsipras et al. 2019; Stutz, Hein, and Schiele 2019). For example, on CIFAR-10, state-of-the-art adversarial training has only about 50% robustness under the strongest adaptive attacks.

One approach to break this robustness bottleneck is to allow rejection of adversarial examples instead of trying to correctly classify all of them. Recently, there has been a great interest in exploring adversarial training of a classifier with a reject option (Laidlaw and Feizi 2019; Stutz, Hein, and Schiele 2020; Sheikholeslami, Lotfi, and Kolter 2021; Pang et al. 2021b; Tramèr 2021). Stutz, Hein, and Schiele proposed to adversarially train confidence-calibrated models so that they can generalize to unseen adversarial attacks. Sheikholeslami, Lotfi, and Kolter modified existing certified-defense mechanisms to allow the classifier to either robustly classify or detect adversarial attacks, and showed that it can lead to better certified robustness, especially for large perturbation sizes. Laidlaw and Feizi proposed a method called Combined Abstention Robustness Learning (CARL) for jointly learning a classifier and the region of the input space on which it should abstain, and showed that training with CARL can result in a more accurate and robust classifier.

## Problem Setup

Let $\mathcal{X} \subseteq \mathbb{R}^d$ denote the space of inputs $\mathbf{x}$ and $\overline{\mathcal{Y}} := \{1, \cdots, k\}$ denote the space of outputs $y$. Let $\mathcal{Y} := \overline{\mathcal{Y}} \cup \{\perp\}$ be the extended output space where $\perp$ denotes the abstain or rejection option. Let $\Delta_k$ denote the set of output probabilities over $\overline{\mathcal{Y}}$ (*i.e.*, the simplex in $k$-dimensions). Let $d(\mathbf{x}, \mathbf{x}')$ be a norm-induced distance on $\mathcal{X}$ (*e.g.*, the $\ell_p$-distance for some $p > 1$), and let $\mathcal{N}(\mathbf{x}, r) := \{\mathbf{x}' \in \mathcal{X} : d(\mathbf{x}', \mathbf{x}) \leq r\}$ denote the neighborhood of $\mathbf{x}$ with distance $r$. Let $\wedge$ and $\vee$ denote the boolean AND and OR operations respectively. Let $\mathbf{1}\{c\}$ define the binary indicator function which takes value 1 (0) when the condition $c$ is true (false). We denote vectors and matrices using boldface symbols.

Given samples from a distribution $\mathcal{D}$ over $\mathcal{X} \times \overline{\mathcal{Y}}$, our goal is to learn a classifier with rejection option, $f : \mathcal{X} \to \mathcal{Y}$, that can correctly classify adversarial examples with small perturbations, and can either correctly classify or reject those with large perturbations. The standard robust error at adversarial budget $\epsilon > 0$ is defined as (Carlini and Wagner 2017)

$$R_\epsilon(f) := \mathop{\mathbb{E}}_{(\mathbf{x},y)\sim\mathcal{D}} \left[ \max_{\mathbf{x}'\in\mathcal{N}(\mathbf{x},\epsilon)} \mathbf{1}\{f(\mathbf{x}') \neq y\} \right],$$

which does not allow any rejection. A few recent works (e.g. (Tramèr 2021)) have proposed a robust error with rejection at adversarial budget $\epsilon > 0$ as

$$R_\epsilon^{\text{rej}}(f) := \mathop{\mathbb{E}}_{(\mathbf{x},y)\sim\mathcal{D}} \Big[ \mathbf{1}\{f(\mathbf{x}) \neq y\}$$
$$\vee \max_{\mathbf{x}'\in\mathcal{N}(\mathbf{x},\epsilon)} \mathbf{1}\{f(\mathbf{x}') \notin \{y, \perp\}\} \Big],$$

which allows the rejection of input perturbations within an $\epsilon$-neighborhood without incurring an error.

Neither of these metrics for robust error is well-suited to our needs. We therefore propose a new metric for evaluating a robust classifier with reject option – the **robust error with rejection** at budgets $\epsilon_0 \in [0, \epsilon]$ and $\epsilon \geq 0$:

$$R_{\epsilon_0,\epsilon}^{\text{rej}}(f) := \mathop{\mathbb{E}}_{(\mathbf{x},y)\sim\mathcal{D}} \Big[ \max_{\mathbf{x}'\in\mathcal{N}(\mathbf{x},\epsilon_0)} \mathbf{1}\{f(\mathbf{x}') \neq y\}$$
$$\vee \max_{\mathbf{x}''\in\mathcal{N}(\mathbf{x},\epsilon)} \mathbf{1}\{f(\mathbf{x}'') \notin \{y, \perp\}\} \Big]. \quad (1)$$

The motivation for this metric is as follows. For small perturbations of a clean input within a neighborhood of radius $\epsilon_0$, both an incorrect prediction and rejection are considered to be an error. For larger perturbations outside the $\epsilon_0$-neighborhood, rejection is not considered to be an error, *i.e.*, the classifier can either predict the correct class or reject larger perturbations.

**Proposition 1.** *The robust error with rejection can be equivalently defined as*

$$R_{\epsilon_0,\epsilon}^{rej}(f) := \mathop{\mathbb{E}}_{(\mathbf{x},y)\sim\mathcal{D}} \Big[ \max_{\mathbf{x}'\in\mathcal{N}(\mathbf{x},\epsilon_0)} \mathbf{1}\{f(\mathbf{x}') = \perp\}$$
$$\vee \max_{\mathbf{x}''\in\mathcal{N}(\mathbf{x},\epsilon)} \mathbf{1}\{f(\mathbf{x}'') \notin \{y, \perp\}\} \Big]. \quad (2)$$

*We first note that*

$$\mathbf{1}\{f(\mathbf{x}') \neq y\} = \mathbf{1}\{f(\mathbf{x}') = \perp\} \vee \mathbf{1}\{f(\mathbf{x}') \notin \{y, \perp\}\}.$$

*The maximum over the $\epsilon_0$-neighborhood can be expressed as*

$$\max_{\mathbf{x}'\in\mathcal{N}(\mathbf{x},\epsilon_0)} \mathbf{1}\{f(\mathbf{x}') \neq y\} = \max_{\mathbf{x}'\in\mathcal{N}(\mathbf{x},\epsilon_0)} \mathbf{1}\{f(\mathbf{x}') = \perp\}$$
$$\vee \max_{\mathbf{x}'\in\mathcal{N}(\mathbf{x},\epsilon_0)} \mathbf{1}\{f(\mathbf{x}') \notin \{y, \perp\}\}.$$

*Finally, the second term in the RHS of the above expression can be combined with the second term inside the expectation of Eq. (1)*, i.e.,

$$\max_{\mathbf{x}'\in\mathcal{N}(\mathbf{x},\epsilon_0)} \mathbf{1}\{f(\mathbf{x}') \notin \{y, \perp\}\} \vee \max_{\mathbf{x}'\in\mathcal{N}(\mathbf{x},\epsilon)} \mathbf{1}\{f(\mathbf{x}') \notin \{y, \perp\}\}$$
$$= \max_{\mathbf{x}'\in\mathcal{N}(\mathbf{x},\epsilon)} \mathbf{1}\{f(\mathbf{x}') \notin \{y, \perp\}\},$$

*which shows the equivalence of (1) and (2).*

Our new metric has the following natural connections with existing metrics in the literature:

- When $\epsilon_0 = \epsilon$, our metric $R^{\text{rej}}_{\epsilon_0,\epsilon}(f)$ reduces to the standard robust error (without rejection) $R_\epsilon(f)$ at budget $\epsilon$.

- When $\epsilon_0 = 0$, our metric reduces to the robust error with rejection at budget $\epsilon$, $R^{\text{rej}}_\epsilon(f)$ proposed *e.g.*, in (Tramèr 2021). For this special case, rejection is considered to be an error *only* for clean inputs (*i.e.*, no perturbation).

- For any classifier $g : \mathcal{X} \to \overline{\mathcal{Y}}$ that does not allow rejection and any $\epsilon_0 \in [0, \epsilon]$, it is easily verified that our metric reduces to the standard robust error at budget $\epsilon$, *i.e.*, $R^{\text{rej}}_{\epsilon_0,\epsilon}(g) = R_\epsilon(g)$.

In our experiments, we evaluate $R^{\text{rej}}_{\epsilon_0,\epsilon}(f)$ over a range of $\epsilon_0$ values by setting $\epsilon_0 = \alpha \epsilon$, $\alpha \in [0, 1]$. This produces a curve with $R^{\text{rej}}_{\alpha\epsilon,\epsilon}(f)$ on the y-axis as a function of $\alpha$ on the x-axis. The curve shows the robust error with rejection of the classifier for a range of small-perturbation neighborhoods, with $R^{\text{rej}}_\epsilon(f)$ at the left end ($\alpha = 0$) and the standard robust error $R_\epsilon(f)$ at the right end ($\alpha = 1$).

## Theoretical Analysis

Our goal is to correctly classify small perturbations of the input and allow rejection of large perturbations when the classifier is not confident. Two fundamental questions arise:

1. *Why not allow rejection of both small and large perturbations?* This is done in most existing studies on robust classification with rejection. On the other hand, many practical applications would like to handle small perturbations, and rejecting them can have severe costs. The question is whether it is possible to correctly classify small perturbations without hurting the robustness *i.e.*, whether it is possible to achieve a small $R^{\text{rej}}_{\epsilon_0,\epsilon}(f)$.

2. *Why not try to correctly classify both small and large perturbations?* This is done in traditional adversarial robustness, typically by adversarial training. The question is essentially about the benefit of allowing rejection.

To answer these questions, we will show that under mild conditions, there exists a classifier $f$ with rejection with a small $R^{\text{rej}}_{\epsilon_0,\epsilon}(f)$. So it is possible to correctly classify small perturbations without rejecting them, answering the first question. Moreover, under the same conditions, all classifiers $g : \mathcal{X} \to \overline{\mathcal{Y}}$ without rejection must have *at least as large* errors, *i.e.*, $R^{\text{rej}}_{\epsilon_0,\epsilon}(g) = R_\epsilon(g) \geq R^{\text{rej}}_{\epsilon_0,\epsilon}(f)$. In fact, the robust error of $g$ may be much larger than that of $f$. This shows the benefit of allowing rejection, answering the second question.

**Theorem 1.** *Consider binary classification. Let $g(\mathbf{x})$ be any decision boundary (*i.e., any classifier without a rejection option). For any $0 \leq \epsilon_0 \leq \epsilon$, there exists a classifier $f$ with a rejection option such that*

$$R^{rej}_{\epsilon_0,\epsilon}(f) \leq R_{(\epsilon_0+\epsilon)/2}(g). \tag{3}$$

*Moreover, the bound is tight: there exist simple data distributions and $g$ such that any $f$ must have $R^{rej}_{\epsilon_0,\epsilon}(f) \geq R_{(\epsilon_0+\epsilon)/2}(g)$.*

The proof for Theorem 1 is given in the Appendix. Intuitively, the theorem states that if the data allows a small robust error at adversarial budget $(\epsilon_0 + \epsilon)/2$, then there exists a classifier with small robust error with rejection at budget $(\epsilon_0, \epsilon)$. For example, if the two classes can be separated with a margin $(\epsilon_0 + \epsilon)/2$, then there exists an $f$ with 0 robust error with rejection, even considering perturbations as large as $\epsilon$ which can be significantly larger than $(\epsilon_0 + \epsilon)/2$. Therefore, under mild conditions, it is possible to classify correctly small perturbations while rejecting large perturbations, answering our first question.

On the other hand, under the same conditions, if we do not allow rejection and consider any classifier $g$ without rejection, then robust error of $g$ at the same adversarial budget is $R^{\text{rej}}_{\epsilon_0,\epsilon}(g) = R_\epsilon(g) \geq R_{(\epsilon_0+\epsilon)/2}(g) \geq R^{\text{rej}}_{\epsilon_0,\epsilon}(f)$. In fact, there can be a big gap between $R_\epsilon(g)$ and $R_{(\epsilon_0+\epsilon)/2}(g)$, *e.g.*, when a large fraction of inputs have distances in $((\epsilon_0 + \epsilon)/2, \epsilon)$ to the decision boundary of $g$. In this case, allowing rejection can bring significant benefit, answering our second question. Note that if we set $\epsilon_0 = 0$, then the theorem reduces to Theorem 5 in (Tramèr 2021).

## Proposed Method

Consider a classifier without rejection $g(\mathbf{x}; \boldsymbol{\theta}_c)$, $g : \mathcal{X} \mapsto \overline{\mathcal{Y}}$ realized by a DNN with parameters $\boldsymbol{\theta}_c$. The output of the DNN is the predicted probability of each class $\mathbf{h}(\mathbf{x}; \boldsymbol{\theta}_c) = [h_1(\mathbf{x}; \boldsymbol{\theta}_c), \cdots, h_k(\mathbf{x}; \boldsymbol{\theta}_c)] \in \Delta_k$. We define the logits or the vector of un-normalized predictions as $\widetilde{\mathbf{h}}(\mathbf{x}; \boldsymbol{\theta}_c) = [\widetilde{h}_1(\mathbf{x}; \boldsymbol{\theta}_c), \cdots, \widetilde{h}_k(\mathbf{x}; \boldsymbol{\theta}_c)] \in \mathbb{R}^k$. The output of the DNN is obtained by applying the softmax function to the logits. The class corresponding to the maximum predicted probability is returned by $g$, *i.e.*, $g(\mathbf{x}; \boldsymbol{\theta}_c) := \operatorname{argmax}_{y \in \overline{\mathcal{Y}}} h_y(\mathbf{x}; \boldsymbol{\theta}_c)$. The corresponding maximum probability is referred to as the *prediction confidence* $h_{\max}(\mathbf{x}; \boldsymbol{\theta}_c) := \max_{y \in \overline{\mathcal{Y}}} h_y(\mathbf{x}; \boldsymbol{\theta}_c)$. The prediction confidence has been used in prior works for determining when the classifier should abstain from prediction (Wu et al. 2018; Stutz, Hein, and Schiele 2020). In this work, we focus on the robust training of a classifier with a confidence-based reject option. Unlike many prior works, the confidence is not simply used at test time for rejection, but is included in our robust training objective.

We define a general classifier with a confidence-based reject option $f : \mathcal{X} \mapsto \mathcal{Y}$ as follows

$$f(\mathbf{x}; \boldsymbol{\theta}) := \begin{cases} g(\mathbf{x}; \boldsymbol{\theta}_c) & \text{if } h_\perp(\mathbf{x}; \boldsymbol{\theta}) \leq \eta, \\ \perp & \text{otherwise}, \end{cases} \tag{4}$$

where $h_\perp(\mathbf{x}; \boldsymbol{\theta}) \in [0, 1]$ is the predicted *probability of rejection* and $\eta \in [0, 1]$ is a suitably-chosen threshold. We can view $h_\perp(\mathbf{x}; \boldsymbol{\theta})$ as a detector that either accepts or rejects an input based on the classifier's prediction, as shown in Fig.1. The detector is defined as a general parametric function of the classifier's un-normalized prediction $h_\perp(\mathbf{x}; \boldsymbol{\theta}) := u(\widetilde{\mathbf{h}}(\mathbf{x}; \boldsymbol{\theta}_c); \boldsymbol{\theta}_d)$, $u : \mathbb{R}^k \mapsto [0, 1]$, with detector-specific parameters $\boldsymbol{\theta}_d$ [1]. Here, we denote the combined parameter vector of the classifier and detector by $\boldsymbol{\theta}^T := [\boldsymbol{\theta}_c^T \, \boldsymbol{\theta}_d^T]$.

---

[1] We discuss specific choices for the function $u$ in the sequel.

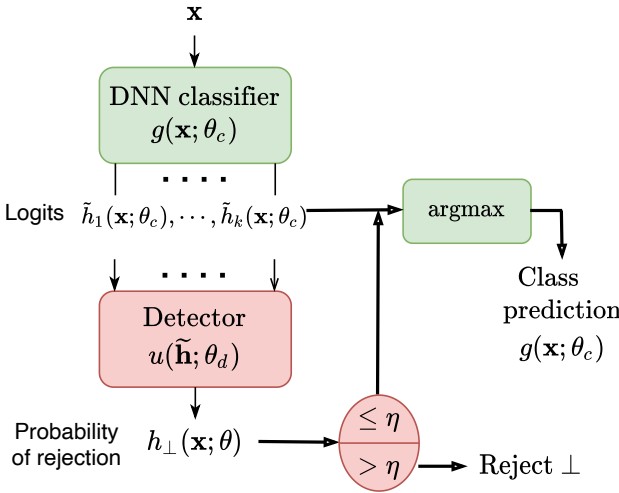

Figure 1: Overview of the proposed classifier with rejection.

**Probability Model.** The class-posterior probability model of the classifier with reject option $f$ is defined as follows:

$$P_{\boldsymbol{\theta}}(y \,|\, \mathbf{x}) = (1 - h_{\perp}(\mathbf{x}\,;\boldsymbol{\theta}))\, h_y(\mathbf{x}\,;\boldsymbol{\theta}_c)\, \mathbf{1}\{y \neq \perp\}$$
$$+ \, h_{\perp}(\mathbf{x}\,;\boldsymbol{\theta})\, \mathbf{1}\{y = \perp\}. \quad (5)$$

An input $\mathbf{x}$ is accepted with probability $1 - h_{\perp}(\mathbf{x}\,;\boldsymbol{\theta})$ and predicted into one of the classes $y \in \overline{\mathcal{Y}}$ with probability $h_y(\mathbf{x}\,;\boldsymbol{\theta}_c)$; otherwise $\mathbf{x}$ is rejected with probability $h_{\perp}(\mathbf{x}\,;\boldsymbol{\theta})$ and the class $\perp$ is returned with probability 1.

## Loss Functions

Consider the robust error with rejection defined in Eq. (1). We would like to design smooth surrogate loss functions to replace the $0 - 1$ error functions in order the minimize the robust error with rejection.

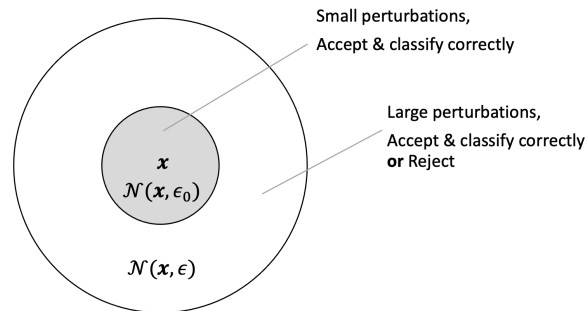

Figure 2: Nested perturbation balls (relative to the $\ell_2$-norm) around a clean input $\mathbf{x}$; used to formalize our robust classification with rejection setting.

**Accept & Classify Correctly.** First, consider the $0 - 1$ error corresponding to small perturbations in the $\epsilon_0$-neighborhood, $\mathbf{1}\{f(\mathbf{x}') \neq y\}$. We would like the corresponding surrogate loss to take a small value when the predicted probability for the true class $y$ is high *and* the predicted probability of rejection is low. The predicted probability of $f$ can be viewed as a $k + 1$ dimensional probability vector over the $k$ classes and the

reject class $\mathcal{Y}$: $[(1 - h_{\perp}(\mathbf{x}'\,;\boldsymbol{\theta}))\, h_1(\mathbf{x}'\,;\boldsymbol{\theta}_c), \cdots, (1 - h_{\perp}(\mathbf{x}'\,;\boldsymbol{\theta}))\, h_k(\mathbf{x}'\,;\boldsymbol{\theta}_c), h_{\perp}(\mathbf{x}'\,;\boldsymbol{\theta})]$. Note that the final term corresponds to the probability of rejection, and the $k + 1$ probabilities sum to 1. For an input $(\mathbf{x}', y)$ to be accepted and predicted into class $y$, the target $k + 1$ dimensional one-hot probability vector has a 1 corresponding to class $y$ and zeros elsewhere. We propose to use the cross-entropy loss between this target one-hot probability and the predicted probability of $f$, given by

$$\ell_{\mathrm{CE}}(\mathbf{x}', y\,;\boldsymbol{\theta}) = -\log[(1 - h_{\perp}(\mathbf{x}'\,;\boldsymbol{\theta}))\, h_y(\mathbf{x}'\,;\boldsymbol{\theta}_c)]. \quad (6)$$

We observe that the above loss function approaches 0 when the probability of rejection is close to 0 and the predicted probability of class $y$ is close to 1; the loss function takes a large value in all other cases. We also apply this cross-entropy loss for clean inputs since we expect the classifier to accept and correctly classify them.

**Accept & Classify Correctly or Reject.** Consider the $0-1$ error corresponding to perturbations in the $\epsilon$-neighborhood, $\mathbf{1}\{f(\mathbf{x}') \notin \{y, \perp\}\}$. We would like the corresponding surrogate loss to take a small value when the predicted probability for the true class is high, or when the probability of rejection is high. To motivate the cross-entropy loss for this case, consider $k$ meta-classes defined as follows: $\{1, \cdots, y \vee \perp, y + 1, \cdots, k\}$, *i.e.*, the reject option is merged only with the true class $y$. The predicted probability of $f$ over these meta-classes is given by: $[(1 - h_{\perp}(\mathbf{x}'\,;\boldsymbol{\theta}))\, h_1(\mathbf{x}'\,;\boldsymbol{\theta}_c), \cdots, (1 - h_{\perp}(\mathbf{x}'\,;\boldsymbol{\theta}))\, h_y(\mathbf{x}'\,;\boldsymbol{\theta}_c) + h_{\perp}(\mathbf{x}'\,;\boldsymbol{\theta}), \cdots, (1 - h_{\perp}(\mathbf{x}'\,;\boldsymbol{\theta}))\, h_k(\mathbf{x}'\,;\boldsymbol{\theta}_c)]$ [2]. For an input $(\mathbf{x}', y)$ to be either *rejected* or *accepted and predicted into class* $y$, the target $k$-dimensional one-hot probability vector has a 1 corresponding to the meta-class $y \vee \perp$, and zeros elsewhere. We propose to use the cross-entropy loss between this target one-hot probability and the predicted probability of $f$, given by

$$\ell_{\mathrm{CE}}^{\mathrm{rej}}(\mathbf{x}', y\,;\boldsymbol{\theta}) = -\log\Big[(1 - h_{\perp}(\mathbf{x}'\,;\boldsymbol{\theta}))\, h_y(\mathbf{x}'\,;\boldsymbol{\theta}_c)$$
$$+ \, h_{\perp}(\mathbf{x}'\,;\boldsymbol{\theta})\Big]. \quad (7)$$

This loss function approaches 0 when i) the probability of rejection is close to 1, **or** ii) the probability of rejection is close to 0 and the predicted probability of class $y$ is close to 1. Note that both the loss functions have a range $[0, \infty)$.

## Robust Training Objective

Given clean labeled samples $(\mathbf{x}, y)$ from a data distribution $\mathcal{D}$, a perturbation budget for robust classification $\epsilon > 0$, and a smaller perturbation budget $\epsilon_0 \in [0, \epsilon]$, we propose the following training objective for learning a robust classifier with a reject option:

$$\mathcal{L}(\boldsymbol{\theta}) = \mathbb{E}_{(\mathbf{x},y)\sim\mathcal{D}}\Big[\ell_{\mathrm{CE}}(\mathbf{x}, y\,;\boldsymbol{\theta}) + \beta \max_{\mathbf{x}' \in \mathcal{N}(\mathbf{x},\epsilon_0)} \ell_{\mathrm{CE}}(\mathbf{x}', y\,;\boldsymbol{\theta})$$
$$+ \, \gamma \max_{\mathbf{x}' \in \mathcal{N}(\mathbf{x},\epsilon)} \ell_{\mathrm{CE}}^{\mathrm{rej}}(\mathbf{x}', y\,;\boldsymbol{\theta})\Big]. \quad (8)$$

---

[2]Note that this is a valid probability distribution over the meta-classes that sums to 1.

The first term corresponds to the standard cross-entropy loss on clean inputs from the data distribution. The second term corresponds to the robust loss for small perturbations in the $\epsilon_0$-neighborhood that we would like the classifier to *accept and correctly classify*. The third term corresponds to the robust loss for large perturbations in the $\epsilon$-neighborhood that we would like the classifier to either *reject* or *accept and correctly classify*. The classifier parameters $\boldsymbol{\theta}_c$ and the detector parameters $\boldsymbol{\theta}_d$ are jointly learned by minimizing $\mathcal{L}(\boldsymbol{\theta})$. The hyper-parameters $\beta \geq 0$ and $\gamma \geq 0$ control the trade-off between the natural error and robust error terms of the classifier. We use standard PGD attack (Madry et al. 2018) to solve the inner maximization in our training objective.

**Comments.** Suppose we choose the detector to always accept inputs, *i.e.*, $h_\perp(\mathbf{x}\,;\boldsymbol{\theta}) = 0$, $\forall \mathbf{x}$, and fix $\epsilon_0 = \epsilon$, $\beta = 1$, $\gamma = 0$, then the training objective specializes to standard adversarial training. The proposed training objective (8) differs from adversarial training by allowing large perturbations of an input to be rejected, when the classifier is likely to predict them incorrectly. As we show experimentally, the proposed method of robust training with rejection typically has higher robustness on unseen adversarial attacks that have a larger perturbation budget $\epsilon$ than that used in training, whereas the robustness of standard adversarial training drops significantly on those unseen adversarial attacks.

## Choice of the Detector

Recall that we defined the detector as a general parametric function of the classifier's un-normalized prediction, that outputs the probability of rejection $h_\perp(\mathbf{x}\,;\boldsymbol{\theta}) = u(\widetilde{\mathbf{h}}(\mathbf{x}\,;\boldsymbol{\theta}_c)\,;\boldsymbol{\theta}_d)$. We explored a few approaches for defining $u(\cdot\,;\boldsymbol{\theta}_d)$ based on smooth approximations of the prediction confidence $\max_{y \in \overline{\mathcal{Y}}} h_y(\mathbf{x}\,;\boldsymbol{\theta}_c)$. For instance, we used a temperature-scaled log-sum-exponential approximation to the $\max$ function, followed by an affine transformation and the Sigmoid function (in order to get a probabilistic output). We also explored a multilayer fully-connected neural network to model the detector, which takes the prediction logits as its input and predicts the probability of rejection. We found the neural network-based model of the detector to have consistently better performance compared to the simple confidence-based approaches. Therefore, we adopt this model of the detector in our experiments.

## Design of Adaptive Attacks

We design strong adaptive attacks to evaluate the robustness with rejection of our method. To compute robustness with rejection at budgets $\epsilon_0$ and $\epsilon$, we need to generate two adversarial examples $\mathbf{x}' \in \mathcal{N}(\mathbf{x}, \epsilon_0)$ and $\mathbf{x}'' \in \mathcal{N}(\mathbf{x}, \epsilon)$ for each clean input $(\mathbf{x}, y)$. We generate the adversarial example $\mathbf{x}'$ within the $\epsilon_0$-ball $\mathcal{N}(\mathbf{x}, \epsilon_0)$ using the following objective:

$$\mathbf{x}' = \operatorname*{argmax}_{\mathbf{x}' \in \mathcal{N}(\mathbf{x}, \epsilon_0)} -\log\left(1 - h_\perp(\mathbf{x}'\,;\boldsymbol{\theta})\right).$$

The goal of the adversary is to make the detector reject the adversarial input by pushing the probability of rejection close

to $1$ [3]. We generate the adversarial example $x''$ within the larger $\epsilon$-ball $\mathcal{N}(\mathbf{x}, \epsilon)$ via the following objective:

$$\mathbf{x}'' = \operatorname*{argmax}_{\mathbf{x}'' \in \mathcal{N}(\mathbf{x}, \epsilon)} \ell_{\text{CE}}^{\text{rej}}(\mathbf{x}'', y\,;\boldsymbol{\theta}).$$

By solving this objective, the adversary attempts to push *both* the probability of rejection $h_\perp(\mathbf{x}''\,;\boldsymbol{\theta})$ and the predicted probability of the true class $h_y(\mathbf{x}''\,;\boldsymbol{\theta}_c)$ close to 0. Thus, the goal of the adversary is to make the classifier-detector accept and incorrectly classify the adversarial input.

We use the Projected Gradient Descent (PGD) method with Backtracking proposed by (Stutz, Hein, and Schiele 2020) to solve the attack objectives. The hyperparameters for PGD with backtracking are specified in the experiment section. Adaptive attacks for evaluating the baseline methods are discussed in the Appendix.

## Experiments

In this section, we perform experiments to evaluate the proposed method (SATR) and compare it to the baseline methods. Our main findings are summarized as follows:

1) SATR achieves higher robustness with rejection compared to adversarial training (with and without confidence-based rejection) and CCAT (Stutz, Hein, and Schiele 2019).

2) On small perturbations, SATR has a much lower rejection rate compared to CCAT, which often rejects a large fraction of the perturbed inputs;

3) Our method outperforms both CCAT and adversarial training under unseen attacks;

We next provide details on the experimental setup, datasets and DNN architectures, baseline methods, and the performance metric.

## Setup

We describe the important experimental settings in this section, and provide additional details about our method and the baselines in the appendix.

**Datasets.** We perform experiments on the MNIST (LeCun 1998) and CIFAR-10 (Krizhevsky, Hinton et al. 2009) image datasets. MNIST contains 50,000 training images and 10,000 test images from 10 classes corresponding to handwritten digits. CIFAR-10 contains 50,000 training images and 10,000 test images from 10 classes corresponding to object categories. Following the setup in (Stutz, Hein, and Schiele 2020), we compute the accuracy of the models on the first 9,000 images of the test set and compute the robustness of the models on the first 1,000 images of the test set. We use the last 1,000 images of the test set as a validation dataset for selecting the rejection threshold of the methods.

---

[3]We appeal to the definition of robust error with rejection in Eq. (2), where rejecting a perturbed input in the $\epsilon_0$-neighborhood is considered an error.

**Baseline Methods.** We compare the performance of SATR with the following three baselines: (1) AT: standard adversarial training without rejection (i.e. accepting every input) (Madry et al. 2018); (2) AT + Rejection: adversarial training with rejection based on the prediction confidence; (3) CCAT: confidence-calibrated adversarial training (Stutz, Hein, and Schiele 2020).

**DNN Architectures.** On MNIST, we use the LeNet network architecture (LeCun et al. 1989) for the classifier $\boldsymbol{\theta}_c$, and use a three-layer fully-connected neural network with width 256 and ReLU activation function for the detector $\boldsymbol{\theta}_d$. On CIFAR-10, we use the ResNet-20 network architecture (He et al. 2016) for the classifier $\boldsymbol{\theta}_c$, and use a seven-layer fully-connected neural network with width 1024, ReLU activation function and a batch normalization layer for the detector $\boldsymbol{\theta}_d$.

**Training Details.** On both MNIST and CIFAR-10, we train the model for 100 epochs with a batch size of 128. We use standard stochastic gradient descent (SGD) starting with a learning rate of 0.1. The learning rate is multiplied by 0.95 after each epoch. We use a momentum of 0.9 and do not use weight decay for SGD. We split each training batch into two sub-batches of equal size, and use the first sub-batch for the first and the third loss terms in our training objective (Eq. (8)), and use the second sub-batch for the second loss term in our training objective. This is similar to the 50% adversarial training strategy used *e.g.*, in (Stutz, Hein, and Schiele 2019). We set the hyper-parameters $\beta = 1$ and $\gamma = 1$ in our training objective without tuning. On MNIST, we train the model from scratch, while on CIFAR-10 we use an adversarially-trained model to initialize the classifier parameters $\boldsymbol{\theta}_c$. On CIFAR-10, we also augment the training images using random crop and random horizontal flip. We use the standard PGD attack (Madry et al. 2018) to generate adversarial training examples. On MNIST, we use the PGD attack with a step size of 0.01, 40 steps and a random start. On CIFAR-10, we use the PGD attack with a step size of $2/255$, 10 steps and a random start. In the training objective, by default, we set $\epsilon = 0.3$ and $\epsilon_0 = 0.1$ for MNIST, and set $\epsilon = 8/255$ and $\epsilon_0 = 2/255$ for CIFAR-10.

**Performance Metric.** We use the *robustness with rejection* at budgets $\epsilon_0$ and $\epsilon$, defined as $1 - R^{\text{rej}}_{\epsilon_0, \epsilon}(f)$, as the evaluation metric. For a fixed $\epsilon$, we vary $\epsilon_0$ from 0 to $\epsilon$ over a given number of values. Note that the $\epsilon_0$ in this performance metric is different from the fixed $\epsilon_0$ that is used in the training objective of the proposed method. For convenience, we define the factor $\alpha := \epsilon_0 / \epsilon \in [0, 1]$, and calculate the robustness with rejection metric for each of the $\alpha$ values from the set $\{0, 0.05, 0.1, 0.2, 0.3, 0.4, 0.5, 1.0\}$. Note that each $\alpha$ value corresponds to an $\epsilon_0$ value equal to $\alpha\epsilon$. We plot a robustness curve for each method, where the $\alpha$ value is plotted on the x-axis and the corresponding robustness with rejection metric is plotted on the y-axis. A larger value of robustness corresponds to better performance. Referring to Fig. 3, we note that at the right end of this curve ($\epsilon_0 = \epsilon$), the robustness $1 - R^{\text{rej}}_{\epsilon, \epsilon}(f)$ corresponds to the standard definition of adversarial robustness without rejection (Madry et al. 2018). At the left end of this curve ($\epsilon_0 = 0$), the robustness

$1 - R^{\text{rej}}_{0, \epsilon}(f)$ corresponds to the robustness with rejection defined by (Tramèr 2021). In practice, we are mainly interested only in the robustness for small values of $\alpha$, where the radius of perturbations (to be accepted) is small.

**Evaluation.** We use the same approach to set the rejection threshold for all the methods. Specifically, on MNIST, we set the threshold such that only 1% of clean validation inputs are rejected. On CIFAR-10, we set the threshold such that only 5% of clean validation inputs are rejected. We consider $\ell_\infty$-norm bounded attacks and generate adversarial examples to compute the robustness with rejection metric via the PGD attack with backtracking (Stutz, Hein, and Schiele 2020). We use a base learning rate of 0.05, momentum factor of 0.9, a learning rate factor of 1.25, 200 iterations, and 10 random restarts for generating adversarial examples $\mathbf{x}'$ within the $\epsilon_0$-ball $\mathcal{N}(\mathbf{x}, \epsilon_0)$. For generating adversarial examples $x''$ within the larger $\epsilon$-ball $\mathcal{N}(\mathbf{x}, \epsilon)$, we use a base learning rate of 0.001, a momentum factor of 0.9, a learning rate factor of 1.1, 1000 iterations, and 10 random restarts.

## Results

We discuss the performance of the proposed method and the baselines on the CIFAR-10 and MNIST datasets.

**Evaluation under seen attacks.** Figure 3 compares the robustness with rejection of the methods as a function of $\alpha$ for the scenario where the adaptive attacks used for evaluation use the same $\epsilon$ budget that was used for training the methods. For the proposed SATR, the $\epsilon_0$ value used for training is indicated (with the corresponding $\alpha$ value) using the vertical dashed line. We observe that CCAT has comparable robustness to AT only for $\alpha = 0$, but its robustness quickly drops for larger $\alpha$. This suggests that CCAT rejects a large fraction of small input perturbations based on its confidence thresholding method. AT with confidence-based rejection has higher robustness compared to standard AT on both datasets, which suggests that including even a simple rejection mechanism can help improve the robustness. On CIFAR-10, the proposed SATR has significantly higher robustness with rejection for small to moderate $\alpha$, and its robustness drops only for large $\alpha$ values (which are not likely to be of practical interest). On MNIST, the robustness of SATR is slightly better than or comparable to AT + Rejection for small to moderate $\alpha$. This suggests that SATR is successful at accepting and correctly classifying a majority of adversarial attacks of small size.

**Evaluation under unseen attacks.** Figure 4 compares the robustness with rejection of the methods as a function of $\alpha$ for the unseen-adaptive-attack scenario, wherein a larger $\epsilon$ (compared to training) is used for evaluation. AT, both with and without rejection, performs poorly in this setting, suggesting that it does not generalize well to unseen (stronger) attacks. CCAT has relatively high robustness for $\alpha = 0$; however, its robustness sharply drops for larger $\alpha$ values. The significantly higher robustness of SATR for a range of small to moderate $\alpha$ values suggests that the proposed training method learns to reject larger input perturbations, even if the attack is unseen.

**Ablation study.** We performed an ablation experiment to study the effect of the hyper-parameter $\epsilon_0$ used by SATR

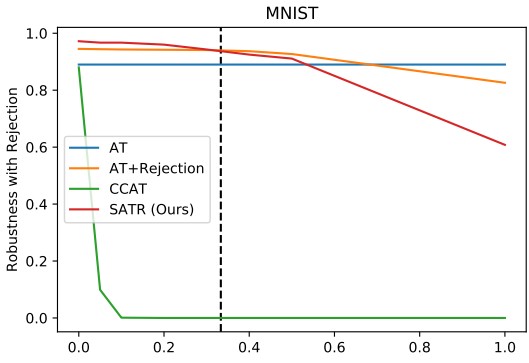
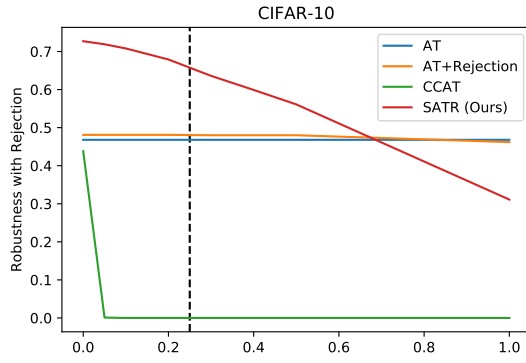

Figure 3: Results on MNIST and CIFAR-10 datasets under **seen** adaptive attacks. On MNIST we set the perturbation budget $\epsilon = 0.3$, while on CIFAR-10 we set the perturbation budget $\epsilon = 8/255$. The vertical dashed line corresponds to the $\epsilon_0 = \alpha\epsilon$ used for training SATR.

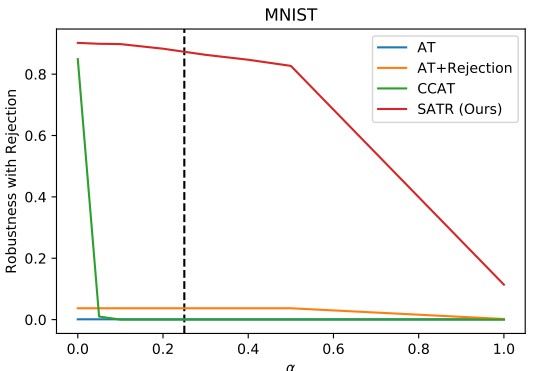
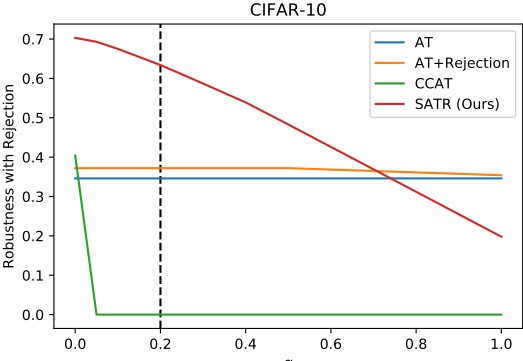

Figure 4: Results on MNIST and CIFAR-10 datasets under **unseen** adaptive attacks. On MNIST, we set the perturbation budget $\epsilon = 0.4$, while on CIFAR-10 we set the perturbation budget $\epsilon = 10/255$. The vertical dashed line corresponds to the $\epsilon_0 = \alpha\epsilon$ used for training SATR.

during training. The result of this experiment is shown in Figure 5 for a few different $\epsilon_0$ values. Clearly, the choice $\epsilon_0 = 0$ leads to poor robustness with rejection, suggesting that a small non-zero value of $\epsilon_0$ is required for training to ensure that SATR does not reject too many small adversarial perturbations. We also observe that a larger $\epsilon_0$ during training typically leads to a higher robustness for large $\alpha$ values. However, this may come at the expense of lower robustness for small $\alpha$, as observed on CIFAR-10 for $\epsilon_0 = 4/255$.

## Conclusion

We explored the problem of learning an adversarially-robust classifier with a reject option. We conducted a careful theoretical analysis of the problem and motivate the need for not rejecting small perturbations of the input. We proposed a novel metric for evaluating the robustness of a classifier with reject option that subsumes prior definitions of robustness, and provides a more fine-grained analysis of the radius (size) of perturbations rejected by a given method. We proposed a novel training objective for learning a robust classifier with rejection that encourages small input perturbations to be accepted and classified correctly, while allowing larger

input perturbations to be rejected when the classifier's prediction may be incorrect. Experimental evaluations using strong adaptive attacks demonstrate significant improvement in the adversarial robustness with rejection of the proposed method, including the setting where unseen attacks with a larger $\epsilon$ budget are present during evaluation.

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

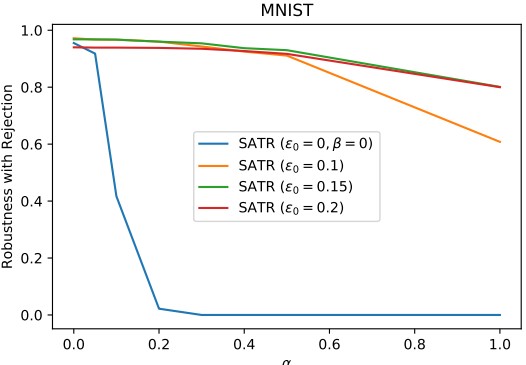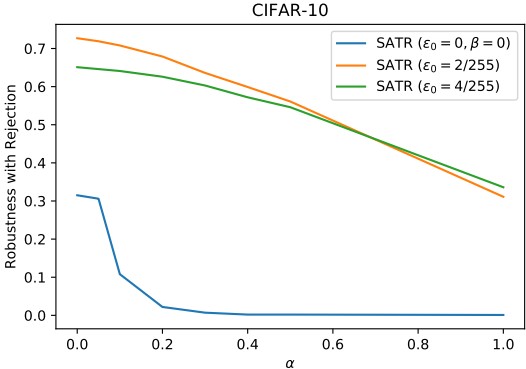

Figure 5: Ablation study (effect of $\epsilon_0$ on SATR) on MNIST and CIFAR-10 datasets under **seen** adaptive attacks. On MNIST we set the perturbation budget $\epsilon = 0.3$, while on CIFAR-10 we set the perturbation budget $\epsilon = 8/255$.

Conference on Machine Learning (ICML), volume 139 of *Proceedings of Machine Learning Research*, 1507–1517. PMLR.

Cortes, C.; DeSalvo, G.; and Mohri, M. 2016. Learning with Rejection. In *Algorithmic Learning Theory - 27th International Conference, ALT*, volume 9925 of *Lecture Notes in Computer Science*, 67–82.

Geifman, Y.; and El-Yaniv, R. 2019. SelectiveNet: A Deep Neural Network with an Integrated Reject Option. In *Proceedings of the 36th International Conference on Machine Learning (ICML)*, volume 97 of *Proceedings of Machine Learning Research*, 2151–2159. PMLR.

Guan, H.; Zhang, Y.; Cheng, H.; and Tang, X. 2018. Abstaining Classification When Error Costs are Unequal and Unknown. *CoRR*, abs/1806.03445.

He, K.; Zhang, X.; Ren, S.; and Sun, J. 2016. Deep Residual Learning for Image Recognition. In *2016 IEEE Conference on Computer Vision and Pattern Recognition, CVPR 2016, Las Vegas, NV, USA, June 27-30, 2016*, 770–778. IEEE Computer Society.

Kato, M.; Cui, Z.; and Fukuhara, Y. 2020. ATRO: Adversarial Training with a Rejection Option. *CoRR*, abs/2010.12905.

Krizhevsky, A.; Hinton, G.; et al. 2009. Learning multiple layers of features from tiny images.

Laidlaw, C.; and Feizi, S. 2019. Playing it Safe: Adversarial Robustness with an Abstain Option. *CoRR*, abs/1911.11253.

LeCun, Y. 1998. The MNIST database of handwritten digits. *http://yann. lecun. com/exdb/mnist/*.

LeCun, Y.; Boser, B. E.; Denker, J. S.; Henderson, D.; Howard, R. E.; Hubbard, W. E.; and Jackel, L. D. 1989. Handwritten Digit Recognition with a Back-Propagation Network. In Touretzky, D. S., ed., *Advances in Neural Information Processing Systems 2, [NIPS Conference, Denver, Colorado, USA, November 27-30, 1989]*, 396–404. Morgan Kaufmann.

Madry, A.; Makelov, A.; Schmidt, L.; Tsipras, D.; and Vladu, A. 2018. Towards Deep Learning Models Resistant to Adversarial Attacks. In *6th International Conference on Learning Representations, Conference Track Proceedings*. OpenReview.net.

Pang, T.; Yang, X.; Dong, Y.; Su, H.; and Zhu, J. 2021a. Bag of Tricks for Adversarial Training. In *9th International Conference on Learning Representations (ICLR)*. OpenReview.net.

Pang, T.; Zhang, H.; He, D.; Dong, Y.; Su, H.; Chen, W.; Zhu, J.; and Liu, T. 2021b. Adversarial Training with Rectified Rejection. *CoRR*, abs/2105.14785.

Sheikholeslami, F.; Lotfi, A.; and Kolter, J. Z. 2021. Provably robust classification of adversarial examples with detection. In *9th International Conference on Learning Representations (ICLR)*. OpenReview.net.

Stutz, D.; Hein, M.; and Schiele, B. 2019. Disentangling Adversarial Robustness and Generalization. In *IEEE Conference on Computer Vision and Pattern Recognition CVPR*, 6976–6987. Computer Vision Foundation / IEEE.

Stutz, D.; Hein, M.; and Schiele, B. 2020. Confidence-Calibrated Adversarial Training: Generalizing to Unseen Attacks. In *Proceedings of the 37th International Conference on Machine Learning (ICML)*, volume 119 of *Proceedings of Machine Learning Research*, 9155–9166. PMLR.

Tax, D. M. J.; and Duin, R. P. W. 2008. Growing a multiclass classifier with a reject option. *Pattern Recognition Letters*, 29(10): 1565–1570.

Tramèr, F. 2021. Detecting Adversarial Examples Is (Nearly) As Hard As Classifying Them. *CoRR*, abs/2107.11630.

Tramèr, F.; Carlini, N.; Brendel, W.; and Madry, A. 2020. On Adaptive Attacks to Adversarial Example Defenses. In *Advances in Neural Information Processing Systems 33: Annual Conference on Neural Information Processing Systems*.

Tsipras, D.; Santurkar, S.; Engstrom, L.; Turner, A.; and Madry, A. 2019. Robustness May Be at Odds with Accuracy. In *7th International Conference on Learning Representations (ICLR)*. OpenReview.net.

Wu, X.; Jang, U.; Chen, J.; Chen, L.; and Jha, S. 2018. Reinforcing Adversarial Robustness using Model Confidence Induced by Adversarial Training. In *Proceedings of the 35th International Conference on Machine Learning (ICML)*, volume 80 of *Proceedings of Machine Learning Research*, 5330–5338. PMLR.

Zhang, H.; Yu, Y.; Jiao, J.; Xing, E. P.; Ghaoui, L. E.; and Jordan, M. I. 2019. Theoretically Principled Trade-off between Robustness and Accuracy. In *Proceedings of the 36th International Conference on Machine Learning (ICML)*, volume 97 of *Proceedings of Machine Learning Research*, 7472–7482. PMLR.

# Appendix

## Proof for Theorem 1

**Theorem 2** (Restatement of Theorem 1). *Consider binary classification. Let $g(\mathbf{x})$ be any decision boundary (i.e., any classifier without a rejection option). For any $0 \leq \epsilon_0 \leq \epsilon$, there exists a classifier $f$ with a rejection option such that*

$$R_{\epsilon_0,\epsilon}^{rej}(f) \leq R_{(\epsilon_0+\epsilon)/2}(g). \tag{9}$$

*Moreover, the bound is tight: there exist simple data distributions and $g$ such that any $f$ must have $R_{\epsilon_0,\epsilon}^{rej}(f) \geq R_{(\epsilon_0+\epsilon)/2}(g)$.*

*Proof.* For any $r > 0$, let $\mathcal{N}(g, r)$ denote the region within distance $r$ to the decision boundary of $g$:

$$\mathcal{N}(g, r) := \{\mathbf{x} \in \mathcal{X} : \exists \mathbf{x}', d(\mathbf{x}', \mathbf{x}) \leq r \text{ and } g(\mathbf{x}') \neq g(\mathbf{x})\}.$$

Consider a parameter $\delta \in [0, \epsilon]$ and construct a classifier $f_\delta$ with rejection as follows:

$$f_\delta(\mathbf{x}) := \begin{cases} \bot & \text{if } \mathbf{x} \in \mathcal{N}(g, \delta), \\ g(\mathbf{x}) & \text{otherwise.} \end{cases} \tag{10}$$

We will show that any sample $(\mathbf{x}, y)$ contributing error to $R_{\epsilon_0,\epsilon}^{rej}(f_\delta)$ must also contribute error to $R_{\epsilon'}(g)$, where $\epsilon' = \max\{\epsilon_0 + \delta, \epsilon - \delta\}$. This will prove that $R_{\epsilon_0,\epsilon}^{rej}(f_\delta) \leq R_{\epsilon'}(g)$, which specializes to $R_{\epsilon_0,\epsilon}^{rej}(f_\delta) \leq R_{(\epsilon_0+\epsilon)/2}(g)$ for the choice $\delta = (\epsilon - \epsilon_0)/2$. Consider the following two cases:

- Consider the first type of error in $R_{\epsilon_0,\epsilon}^{rej}(f_\delta)$: $\max_{\mathbf{x}' \in \mathcal{N}(\mathbf{x}, \epsilon_0)} \mathbf{1}\left[f_\delta(\mathbf{x}') \neq y\right] = 1$. This implies that there exists $\mathbf{x}' \in \mathcal{N}(\mathbf{x}, \epsilon_0)$ such that $f_\delta(\mathbf{x}') \neq y$. So there are two subcases to consider:
  (1) $\mathbf{x}' \in \mathcal{N}(g, \delta)$: in this case $\mathbf{x} \in \mathcal{N}(g, \delta + \epsilon_0)$.
  (2) $g(\mathbf{x}') \neq y$: in this case either $g(\mathbf{x}) \neq y$, or $g(\mathbf{x}) = y \neq g(\mathbf{x}')$ and thus $\mathbf{x} \in \mathcal{N}(g, \epsilon_0)$.
  In summary, either $g(\mathbf{x}) \neq y$ or $\mathbf{x} \in \mathcal{N}(g, \epsilon_0 + \delta)$.

- Next consider the second type of error in $R_{\epsilon_0,\epsilon}^{rej}(f_\delta)$: $\max_{\mathbf{x}'' \in \mathcal{N}(\mathbf{x}, \epsilon)} \mathbf{1}\left[f_\delta(\mathbf{x}'') \notin \{y, \bot\}\right] = 1$. This means there exists an $\mathbf{x}'' \in \mathcal{N}(\mathbf{x}, \epsilon)$ such that $f_\delta(\mathbf{x}'') \notin \{y, \bot\}$, i.e., $\mathbf{x}'' \notin \mathcal{N}(g, \delta)$ and $g(\mathbf{x}'') \neq y$. This implies that *all* $\mathbf{z} \in \mathcal{N}(\mathbf{x}'', \delta)$ should have $g(\mathbf{z}) = g(\mathbf{x}'') \neq y$. In particular, there exists $\mathbf{z} \in \mathcal{N}(\mathbf{x}'', \delta)$ with $d(\mathbf{z}, \mathbf{x}) \leq \epsilon - \delta$ and $g(\mathbf{z}) \neq y$. It can be verified that $\mathbf{z} = \frac{\delta}{\epsilon}\mathbf{x} + \frac{\epsilon - \delta}{\epsilon}\mathbf{x}''$, which is a point on the line joining $\mathbf{x}$ and $\mathbf{x}''$, satisfies the above condition. In summary, either $g(\mathbf{x}) \neq y$, or $g(\mathbf{x}) = y \neq g(\mathbf{z})$ and thus $\mathbf{x} \in \mathcal{N}(g, \epsilon - \delta)$.

Overall, a sample $(\mathbf{x}, y)$ contributing error to $R_{\epsilon_0,\epsilon}^{rej}(f_\delta)$ must satisfy either $g(\mathbf{x}) \neq y$ or $\mathbf{x} \in \mathcal{N}(g, \epsilon')$, where $\epsilon' = \max\{\epsilon_0 + \delta, \epsilon - \delta\}$. Clearly, such a sample also contributes an error to $R_{\epsilon'}(g)$. Therefore, we have

$$R_{\epsilon_0,\epsilon}^{rej}(f_\delta) \leq R_{\epsilon'}(g), \tag{11}$$

which leads to the desired bound when $\delta = (\epsilon - \epsilon_0)/2$ and $\epsilon' = (\epsilon + \epsilon_0)/2$.

To show that the bound is tight, consider the following data distribution. Let $\mathbf{x} \in \mathbb{R}$ and $y \in \{-1, +1\}$, $0 < \epsilon_0 < \epsilon$, and let $\alpha \in (0, 1/2)$ be some constant: $(\mathbf{x}, y)$ is $(-4\epsilon, -1)$ with probability $(1 - \alpha)/2$, $(-\epsilon_0/4, -1)$ with probability $\alpha/2$, $(\epsilon_0/4, +1)$ with probability $\alpha/2$, and $(4\epsilon, +1)$ with probability $(1 - \alpha)/2$. Let $g(\mathbf{x}) := \text{sign}(\mathbf{x} + \epsilon)$. It is clear that $R_{(\epsilon_0+\epsilon)/2}(g) = \alpha/2$. It is also clear that any $f$ must have $R_{\epsilon_0,\epsilon}^{rej}(f) \geq \alpha/2$ since the points $\mathbf{x} = -\epsilon_0/4$ and $\mathbf{x} = \epsilon_0/4$ have distance only $\epsilon_0/2$ but have different labels. $\square$

## Experimental Details

### General Setup

**Software and Hardware.** We ran all our experiments with PyTorch and NVIDIA GeForce RTX 2080Ti GPUs.

**Number of Evaluation Runs.** We ran all experiments once with fixed random seeds.

**Dataset.** MNIST (LeCun 1998) is a large dataset of handwritten digits. Each digit has 5,500 training images and 1,000 test images. Each image is 28×28 grayscale. CIFAR-10 (Krizhevsky, Hinton et al. 2009) is a dataset of 32×32 color images with ten classes, each consisting of 5,000 training images and 1,000 test images. The classes correspond to different objects such as dogs, frogs, ships, trucks, etc. We normalize the range of pixel values to [0,1].

**Multiple restart PGD Attacks.** We use PGD attacks with multiple restarts for evaluating the robustness. Following (Stutz, Hein, and Schiele 2020), we initialize the perturbation $\delta$ uniformly over directions and norm:

$$\boldsymbol{\delta} = u\,\epsilon\,\frac{\boldsymbol{\delta}'}{\|\boldsymbol{\delta}'\|_\infty}, \; \boldsymbol{\delta}' \sim \mathcal{N}(\mathbf{0}, \mathbf{I}), \; u \sim U(0,1)$$

where $\boldsymbol{\delta}'$ is sampled from the standard Gaussian and $u \in [0, 1]$ from a uniform distribution. We also consider zero initialization, i.e., $\delta = 0$. We allocate one restart for zero initialization, and multiple restarts for the random initialization. We finally select the perturbation corresponding to the best objective value obtained throughout the optimization.

### Baselines

We consdier three baselines: (1) AT: adversarial training without rejection (i.e. accepting every input); (2) AT+Rejection: adversarial training with confidence-based rejection; (3) CCAT: confidence-calibrated adversarial training. We provide their training details below.

**AT and AT+Rejection.** We consider the standard adversarial training proposed in (Madry et al. 2018) and use the following training objective:

$$\mathcal{L}(\boldsymbol{\theta}) = \mathbb{E}_{(\mathbf{x},y)\sim\mathcal{D}}\left[\ell_{\text{CE}}(\mathbf{x}, y\,; \boldsymbol{\theta}) + \max_{\mathbf{x}' \in \mathcal{N}(\mathbf{x}, \epsilon)} \ell_{\text{CE}}(\mathbf{x}', y\,; \boldsymbol{\theta})\right].$$

We train on 50% clean and 50% adversarial examples per batch. On MNIST, we use the LetNet network architecture (LeCun et al. 1989) and train the network for 100 epochs with a batch size of 128. We use standard stochastic gradient descent (SGD) starting with a learning rate of 0.1. The

learning rate is multiplied by 0.95 after each epoch. We use a momentum of 0.9 and do not use weight decay for SGD. We use the PGD attack to generate adversarial training examples with $\epsilon = 0.3$, a step size of 0.01, 40 steps and a random start. On CIFAR-10, we use the ResNet-20 network architecture (He et al. 2016) and train the network following the suggestions in (Pang et al. 2021a). Specifically, we train the network for 110 epochs with a batch size of 128 using stochastic gradient decent (SGD) with Nesterov momentum and learning rate schedule. We set momentum 0.9 and $\ell_2$ weight decay with a coefficient of $5 \times 10^{-4}$. The initial learning rate is 0.1 and it decreases by 0.1 at 100 and 105 epoch respectively. We augment the training images using random crop and random horizontal flip. We use the PGD attack to generate adversarial training examples with $\epsilon = 8/255$, a step size of $2/255$, 10 steps and a random start.

**CCAT.** We follow the original training settings for CCAT in (Stutz, Hein, and Schiele 2020) and train models on MNIST and CIFAR-10 using standard SGD. On MNIST, we use the LetNet network architecture (LeCun et al. 1989) and train the network for 100 epochs with a batch size of 100 and a learning rate of 0.1. On CIFAR-10, we use the ResNet-20 network architecture (He et al. 2016) and train the network for 200 epochs with a batch size of 100 and a learning rate of 0.075. We augment the training images using random crop and random horizontal flip on CIFAR-10. On both MNIST and CIFAR-10, we use learning rate schedule and the learning rate is multiplied by 0.95 after each epoch. We use a momentum of 0.9 and do not use weight decay for SGD. We use the PGD attack with backtracking to generate adversarial training examples: we use a learning rate of 0.005, a momentum of 0.9, a learning rate factor of 1.5, 40 steps and a random start. We randomly switch between the random initialization and zero initialization. We train on 50% clean and 50% adversarial examples per batch.

### Adaptive Attacks for Confidence-based Detectors

We design adaptive attacks to evaluate the robustness with rejection of classifiers using confidence (or maximum softmax score) to reject adversarial inputs (e.g. AT+Rejection and CCAT). To compute robustness with rejection at budgets $\epsilon_0$ and $\epsilon$, we need to generate two adversarial examples $\mathbf{x}' \in \mathcal{N}(\mathbf{x}, \epsilon_0)$ and $\mathbf{x}'' \in \mathcal{N}(\mathbf{x}, \epsilon)$ for each clean input $(\mathbf{x}, y)$. We generate the adversarial example $\mathbf{x}'$ within the $\epsilon_0$-ball $\mathcal{N}(\mathbf{x}, \epsilon_0)$ using the following objective:

$$\mathbf{x}' = \operatorname*{argmax}_{\mathbf{x}' \in \mathcal{N}(\mathbf{x}, \epsilon_0)} -\sum_{j=1}^{k} h_j(\mathbf{x}'; \boldsymbol{\theta}_c)^2.$$

The goal of the adversary is to make the detector reject the adversarial input by pushing the softmax output of the network close to uniform.

We generate the adversarial example $x''$ within the larger $\epsilon$-ball $\mathcal{N}(\mathbf{x}, \epsilon)$ via the following objective:

$$\mathbf{x}'' = \operatorname*{argmax}_{\mathbf{x}' \in \mathcal{N}(\mathbf{x}, \epsilon)} \max_{j \neq y} h_j(\mathbf{x}'; \boldsymbol{\theta}_c).$$

By solving this objective, the adversary attempts to find misclassified adversarial examples with high confidence. Thus, the goal of the adversary is to make the classifier-detector system accept and incorrectly classify the adversarial input.

We use the Projected Gradient Descent (PGD) method with Backtracking proposed by (Stutz, Hein, and Schiele 2020) to solve the attack objectives. The hyperparameters for PGD with backtracking are specified in the experiment section of the main paper.

### Additional Results

**Evaluation on clean test inputs.** We evaluate our method and the baselines on clean test inputs. We calculate the accuracy with rejection defined as the accuracy on accepted test inputs, and the rejection rate defined as the fraction of rejected test inputs. The results given in Table 1 show that our method SATR has comparable performance to the baselines on clean test inputs.

| Dataset | Method | Acc. with Rej. | Rej. Rate |
|---------|--------|----------------|-----------|
| **MNIST** | AT | 99.08 | 0.00 |
| | AT+Rejection | 99.68 | 1.71 |
| | CCAT | 99.88 | 1.51 |
| | SATR (Ours) | 99.76 | 1.61 |
| **CIFAR-10** | AT | 88.07 | 0.00 |
| | AT+Rejection | 90.73 | 5.42 |
| | CCAT | 91.78 | 5.80 |
| | SATR (Ours) | 91.51 | 4.86 |

Table 1: Evaluation on clean test inputs. The accuracy with rejection is defined as the accuracy on the accepted test inputs and the rejection rate is defined as the fraction of test inputs that are rejected. All values are in percentages.