# OpenReview forum: "Revisiting Adversarial Robustness of Classifiers With a Reject Option"
_AAAI.org/2022/Workshop/AdvML — AAAI-22 AdvML Workshop Oral_

### Official Review · Reviewer_rAKF · 2021-11-30
**Review of Paper 18**

**Rating:** 8
**Confidence:** 4

**Review:**

This paper proposes a metric of robust error with rejection, with the motivation that: 'For small perturbations within $\epsilon_{0}$, both an incorrect prediction and rejection are considered to be an error. For perturbations larger than $\epsilon$, rejection is not considered to be an error'. Based on this new metric, a corresponding adversarial training objective is designed in Equation (8). Empirical evaluations are done on MNIST and CIFAR-10, under both seen and unseen adaptive attacks.

The key to make this paper be widely accepted is to make the community buy in the new definition of *robust error with rejection*, in which the rejection is justified or not based on the magnitude of perturbations. This assumption is somewhat strong, since universal $\epsilon_{0}$ and $\epsilon$ may be suboptimal, e.g., an easy example with large perturbation should still be correctly classified and not be rejected, while similarly a hard example could be assigned a smaller $\epsilon_{0}$. The authors may polish their definition into instance-dependent version, which could be more flexible and reasonable.

---

### Decision · Program_Chairs · 2021-12-01

**Decision:**

Accept (Oral)

**Comment:**

The reviewer recommends acceptance of this paper with an oral presentation.